# Women with Cervical High-Risk Human Papillomavirus: Be Aware of Your Anus! The ANGY Cross-Sectional Clinical Study

**DOI:** 10.3390/cancers14205096

**Published:** 2022-10-18

**Authors:** Martine Jacot-Guillarmod, Vincent Balaya, Jérôme Mathis, Martin Hübner, Fabian Grass, Matthias Cavassini, Christine Sempoux, Patrice Mathevet, Basile Pache

**Affiliations:** 1Gynecology Department, Department Women-Mother-Child, Lausanne University Hospital (CHUV), 1011 Lausanne, Switzerland; 2Faculty of Biology and Medicine (FBM), University of Lausanne, 1005 Lausanne, Switzerland; 3Department of Gynecology and Obstetrics, Foch Hospital, 92150 Suresnes, France; 4Gynecology and Obstetrics Department, Biel Hospital Center, 2501 Biel, Switzerland; 5Department of Obstetrics and Gynecology, University Hospital of Berne and University of Berne, 3012 Berne, Switzerland; 6Department of Visceral Surgery, Lausanne University Hospital (CHUV), 1011 Lausanne, Switzerland; 7Division of Infectious Diseases, Lausanne University Hospital (CHUV), 1011 Lausanne, Switzerland; 8Institute of Pathology, Lausanne University Hospital (CHUV), 1011 Lausanne, Switzerland

**Keywords:** HPV testing, gynecology, proctology, human papillomavirus infections, dysplasia, cervix, anus, screening, cancer

## Abstract

**Simple Summary:**

High-risk HPV (HR-HPV) infection is an established risk factor of cervical cancer. Cases of anal cancer are increasingly observed in women. Since anal infection by HR-HPV is also a risk factor for anal cancer, the aim of this study was to investigate the relation between cervical and anal HPV and dysplasia, as their interaction remains unclear. The present study identified anal HR-HPV as an independent risk factor for patients with cervical HR-HPV (OR 3.3, CI 1.2–9.0, *p* = 0.02). This finding emphasizes the importance of concomitant screening of the anal region in case of cervical HR-HPV.

**Abstract:**

Anogenital human papillomaviruses (HPV) are highly prevalent in sexually active populations, with HR-HPV being associated with dysplasia and cancers. The consequences of cervical HPV infection are well-known, whereas those of the anus are less clear. The correlation of cervical and anal HPVs with the increasing number of anal cancers in women has not been studied yet. The objective of our prospective study was to determine whether cervical and anal HPV correlated in a cohort of women recruited in a university hospital in Switzerland. Recruitment was conducted in the gynecology clinic, the colposcopy clinic, and the HIV clinic. Cervical and anal HPV genotyping and cytology were performed. Overall, 275 patients were included (360 were initially planned), and among them, 102 (37%) had cervical HR-HPV. Patients with cervical HR-HPV compared to patients without cervical HR-HPV were significantly younger (39 vs. 44 yrs, *p* < 0.001), had earlier sexual intercourse (17.2 vs. 18.3 yrs, *p* < 0.01), had more sexual partners (2.9 vs. 2.2, *p* < 0.0001), more dysplastic cervical cytology findings (42% vs. 19%, *p* < 0.0001) and higher prevalence of anal HR-HPV (59% vs. 24%, *p* < 0.0001). Furthermore, the HR-HPV group reported more anal intercourse (44% vs. 29%, *p* < 0.015). Multivariate analysis retained anal HR-HPV as independent risk factor for cervical HR-HPV (OR3.3, CI 1.2–9.0, *p* = 0.02). The results of this study emphasize that it is of upmost importance to screen women for anal HR-HPV when diagnosing cervical HR-HPV.

## 1. Introduction

Human papillomaviruses (HPV) are naked, double-stranded DNA viruses that rely on epithelium differentiation for productive infections. More than 200 different genotypes infect humans, having a tropism for specific epithelia such as skin, oral, and anogenital mucosa. Upon infection of the basal cells, viral DNA replication is tightly controlled by the host cell as well as viral factors. Viral production is intimately linked to terminal differentiation of the infected epithelial cell so that virions are produced only in the upper layers of the infected epithelium. Most infections clear spontaneously under immunological host control. However, long-term persistence over years can be observed. In those cases, cell host alteration and viral oncogene expression can lead to cellular abnormalities, and potentially to cancer [1].

Anogenital HPVs are mainly sexually transmitted. They can infect various epithelia such as the vulva, vagina, cervix, penis, oropharynx, and anus. Specific HPV genotypes considered as high-risk (HR-HPV) are a necessary cause of most of cervical cancers [2]. The same HR-HPV (such as HPV16 or HPV18) that play a major role in cervical cancer are also associated with severe diseases of the anal sphere. The concomitant HPV infection of the cervix and the anogenital region has been partially studied in some populations. For instance, Hawaiian women had concurrent anal and cervical HPV infections in 13% of cases [3,4]. 

HIV is a major risk factor of recurrent HPV infection. Other risks such as tobacco use, men having sex with men (MSM), early onset of sexual activity, or multiple sexual partners have also been identified [5]. Cervical cancer carries a significant burden in terms of individual women’s health concern, but also societal costs (screening, follow-up, therapy, anxiety, years of life lost) [6]. Furthermore, the incidence of anal cancer has increased by 2% per year since the 1970’s in the general population, especially in developed countries. This cancer affects women more than men and is related to HR-HPV in 88% of cases [7,8,9]. To date, routine screening for anal HPV, dysplasia, and cancer is most often limited to HIV-infected patients and MSM patients. In contrast, women with cervical HR-HPV are not included in the recommendations for systematic screening.

The aim of the present study was to determine if patients with cervical HR-HPV infection had concomitant infection of the anus with HR-HPV as well.

## 2. Materials and Methods

### 2.1. Study Setting, Participants, and Sampling

The study was named *ANGY* for *AnusGynecology*. This prospective, cross-sectional, single-center study included women recruited at the Lausanne University Hospital (CHUV), a tertiary university hospital in Switzerland. Patients to be classified either in the cervical HR-HPV-positive group or cervical HR-HPV-negative group were recruited in three outpatient clinics: (1) gynecology clinic (patients with regular problems and follow-up); (2) specialized colposcopy clinic; and (3) HIV clinic. Total number of patients to include was 360, 120 from each of the three outpatient clinics. Based on the prevalence of anal dysplasia in each of the three groups (90% power, significant difference at the two-sided 0.05 level), a sample size of 330 patients was calculated. Considering 10% margin error, the final sample size was set at 360 patients.

Cervical and anal screening of HPV genotypes and cytology were performed by board-certified gynecologists and colorectal surgeons. The operation was completed during a single appointment. A cervical cytology smear (for PAP-test and HPV test), with biopsies if needed (according to the international standards of care in colposcopy), was conducted after vaginal examination and colposcopy on a gynecological chair. The proctology team conducted a proctology examination with high-resolution anoscopy (HRA), along with cytological smears of the anal canal (with biopsies if lesions were visualized). Using a dedicated brush, ectocervical, endocervical, and anal samples were performed. 

HPV tests were performed with the MagNaPure 96 Total Nucleic Acids kit (Roche, Basel, Switzerland), with genotyping of 28 HPV genotypes (6, 11, 16, 18, 26, 31, 33, 35, 39, 40, 42–45, 51–54, 56, 58, 59, 61, 66, 68–70, 73, and 82) by the Anyplex™ II HPV28 kit (Seegene, Seoul, South Korea).

Furthermore, all patients were screened for HIV. A self-assessed questionnaire exploring sexual habits and HPV immunization status was submitted to patients as well (Appendix A). Following variables were collected: demographics and patient characteristics, cervical and anal clinical findings on colposcopy/proctology exam after acetic acid (5%) and Lugol’s iodine solution application, cytology and HPV genotype of both cervical and anal area, HIV serology, and CD4 count. Biopsies were performed at the clinicians’ discretion when clinical cervix/anal examination were abnormal. High-risk HPV genotypes were as follows: 16-18-31-33-45-52-58.

The study was approved by the Institutional Review Board of “Commission Cantonale d’éthique Vaud” (CER-VD # 2015-00200, date of approval: 2015). Informed consent was obtained from all subjects involved in the study.

Recruitment was conducted between September 2016 and October 2020, with premature study discontinuation of the clinical trial due to difficulties in recruiting patients and related to COVID-19 epidemic-related restrictions. Statistical power for the main outcome, cervical HR-HPV, was not impacted in the present statistics due to its high prevalence.

### 2.2. Statistical Analysis

Patients were divided into two groups according to their cervical HR-HPV: positive or negative. The chi-squared test was applied to compare qualitative variables that were expressed as n (percent). The ANOVA test was used to compare quantitative variables that were reported as mean ± standard deviation (SD). *p* values lower than 0.05 were considered as significant. Variables yielding *p* values lower than 0.05 by univariate analysis were entered into a multivariate logistic regression model to determine variables independently associated with the presence of cervical HR-HPV. All statistical analyses were carried out using XLstat Biomed software (AddInsoft V19.4, Paris, France).

## 3. Results

Overall, 275 participants were enrolled until premature study discontinuation. Patient demographics are presented in Table 1. 

Univariate analysis of cytopathologic factors associated with cervical HR-HPV is shown in Table 2.

Table 3 shows the correlation between cervical and anal HPV status for participants without cervical HPV and HR cervical HPV and without anal HPV and HR anal HPV.

Multivariate analysis retained age (OR 0.96, IC 0.92–0.99, *p*-value = 0.02), smoking (OR 2.53, IC 1.08–5.94, *p*-value = 0.03), cervical LR-HPV (OR 3.23, IC 1.30–8.01, *p*-value = 0.01), and anal HR-HPV (OR 2.47, IC 1.01–6.06, *p*-value = 0.048) as factors independently associated with cervical HR-HPV (Appendix A). 

## 4. Discussion

The present study focusing on risk factors for cervical high-risk HPV showed interesting results, with validation of the cohort through demographic results allowing for a comparison with the general HR-HPV population reported in other studies described hereafter. 

This study revealed anal high-risk HPV as independent risk factor for concomitant cervical HR-HPV infection. In patients with cervical HR-HPV, screening of the anal region should be strongly considered. Studies on anal HPV prevalence in women with a history of cervical CIN2+/HSIL are scarce, with a wide variety of numbers ranging from 4% to 86%, depending on the subgroup analyzed. On the other hand, the identified incidence of histological anal high-grade squamous intraepithelial lesions was 0–9% in women with cervical dysplasia [10]. Our study found similar results in line with the prevalence reported in the literature. Of note, no cancers were found in our cohort.

HIV is a recognized risk factor for cervical dysplasia [11]. Evidence for an association with anal dysplasia is growing as well, with guidelines encouraging systematic screening of HIV patients, even if asymptomatic, for anal and cervical HPV/dysplasia [12]. The prevalence of HR-HPV in the HIV population is high, with subsequent higher risk of cancer [13]. For instance, through their systematic review, Lin et al. found out that the prevalence of anal HR-HPV was higher in patients with cervical HR-HPV than in HIV-positive patients, with a risk of having the same HR genotype present in the two spheres [11]. Interestingly, HIV was not retained as risk factor of HPV/dysplasia in our study, most probably due to the smaller sample size (type II error), although we specifically included one arm with HIV-positive patients. This may be explained by the fact that all the HIV patients had undetectable HIV viremia.

As previously stated, cervical cancer is a frequent disease and is mostly caused by HPV. New screening recommendations to decrease the incidence of cervical cancer may thus have a large beneficial effect for the population and the individuals. Our results support the fact that guidelines on cervical cancer screening should include meticulous anal examination with optional cytology ± HPV testing when diagnosing cervical high-grade dysplasia. The data presented in this current study raised the question of a multidisciplinary approach between involved specialties, such as gynecologists and proctologists, to provide proper follow-up. The rational for systematic screening of anal HPV and dysplasia in the general population outside risk populations (HIV, MSM) is debated, as the prevalence of cancer is low [9]. The appropriate screening method is matter of debate, with either genotyping or cytology sampling (or both). Although technically easy screening methods, their cost-effectiveness has not been assessed in large trials yet. However, at-risk groups such as HIV-positive patients, immunosuppressed patients, or patients with HR-HPV in the genital area should be systematically screened [14]. Table 3 shows that when cervical HR-HPV was found, anal HR-HPV was present in 59.3% of cases. On the other hand, when no cervical HPV was detected, there was no anal HPV in 66.2% of participants This result emphasizes the importance of a suspicious attitude towards concomitant presence of anal HPV when cervical HPV is present.

Younger age was identified as an independent risk factor for cervical HR-HPV. As we know to date, there is a clearing of HPV in human cells over time, with some genotypes being more carcinogenic. Young patients tend to have a higher viral load but also different concomitant subtypes of genotypes [15].

Low-risk HPV was retained as risk factor for cervical HR-HPV. This is in line with the literature, as stated by Cogliano et al., along with the IARC classification of HPV genotypes (https://monographs.iarc.who.int/wp-content/uploads/2018/06/mono90.pdf (accessed on 17 October 2022)), stating that HR-HPVs, when they are found, are more often present with LR-HPVs. These findings support the hypothesis of coinfection/cohabitation models [2].

The detrimental role of smoking has been established in multiple cancers, and both the cervix and the anus are no exception. Our results are congruent with those in the literature. Cervical cancer represents a significant burden in the population, not only in terms of number of cancer incidences, but also from the economic perspective [6]. From an epidemiological point of view, our results support the important role that health care stakeholders and politics play in preventing tobacco use. 

Interestingly, in our study, the number of sexual partners was not retained as an independent risk factor for cervical HR-HPV, although it is described in the literature [5]. One could reasonably assume that there is a relationship between exposure to HPV through sexual intercourse, but this was not observed in our population. One reason may be a potential role of exposure time rather than repeated exposure as an oncogenic explanation. The distribution of genotype may reflect sexual activity according to age.

The results of this study support the promotion of HPV vaccination of women. Interestingly, vaccination against HPV had no effect in our cohort. This is most probably due to the limited number of patients and the rather high mean age of the cohort. Furthermore, the vaccination rate was very low. This highlights the real effects of the disease in the unvaccinated population. With national programs of vaccination, which have been implemented worldwide (but especially in industrialized countries), we should expect to see lower rates of HPV in women and decreasing cancer cases [16,17,18]. As HPV involved in cervical cancer also plays a role in anal cancer, a ripple effect on anal HPV with a decreased incidence is to be expected [11].

It could be very interesting to follow up with our participants, in order to observe the evolution of HPV prevalence over time, and the eventual development of cancers. On the opposite, one should not forget the natural course of HPV infection, with most of the viral load being cleared from cervical and anal cells as time passes [2]. 

Several limitations of the present study need to be addressed. There is an inherent weakness due to the cross-sectional design of the study. For instance, the difficulty to reliably assess both HPV and dysplasia impedes the identification of a causal inference. However, as highlighted by Wang et al., the design is useful to establish preliminary evidence before planning future large-scale studies [19]. The rather low sample size requires careful interpretation of the results. The comparison and study of each HPV genotype was not performed due to the small number of patients included. The extended recruitment time might also be a source of bias due to staff changes. We tried to address this limitation through continuous education and training of teams with the recruitment and procedures methods. One pitfall is the premature termination of the study, due to lack of recruitment over time and restrictions in recruitment with the COVID-19 epidemic. Unfortunately, it is not an exception, considering that about one-quarter of planned RCTs are prematurely discontinued. Research data should be made available for further research [20], especially in light of the results, in order to contribute to future larger studies and thus enrich scientific literature. It can also be considered as a waste of scarce resources, a loss of valuable research data, and most importantly, missed opportunities to learn from failures.

In our experience, anal screening during a gynecologic exam is not acceptable for many women. The results of this study may help to increase the acceptance of anal screening.

The strength of the present study is the prospective collection of concomitant samples from both the anus and the cervix, not only in patients with known cervical dysplasia, but also in a comparative group without any lesions. This study supports the assumption, as suggested by Acevedo-Fontánez et al., that women with gynecological HPV-related malignancies are more likely to develop secondary anal cancer than women in the general population [21].

## 5. Conclusions

In conclusion, genital HPV is highly prevalent in women, and high-risk genotypes should be closely monitored in order to prevent cervical and anal dysplasia as well as cancer development. Most importantly, when cervical HR- HPV is diagnosed, anal screening for concomitant HPV or dysplasia should be proposed, as there can be dramatic consequences for the health of our patients if neglected. 

## Figures and Tables

**Table 1 cancers-14-05096-t001:** Demographics of total population, patients without cervical HR-HPV, and patients with cervical HR-HPV.

Demographic and Anamnesis	Total Population N = 275		Cervical HR-HPV-Negative N = 173		Cervical HR-HPV-Positive N = 102		*p*
	n Mean ± SD	[%] [Range]	n Mean ± SD	[%] [Range]	n Mean ± SD	[%] [Range]	
Age [years] (mean)	42.2 ± 12.0	[20–78]	44.0 ± 11.8	[20–78]	39.0 ± 11.6	[22–69]	**0.001**
Gestity (mean)	2.2 ± 2.1	[0–16]	2.4 ± 1.8	[0–10]	2.0 ± 2.5	[0–16]	0.18
Parity (mean)	2.8 ± 1.5	[0–5]	2.6 ± 1.5	[0–5]	3.1 ± 1.6	[1–5]	**0.03**
Education							
Secondary school	66	25.6	45	27.8	21	21.9	0.31
Apprenticeship	68	26.4	45	27.8	23	24.0	
High school	32	12.4	22	13.6	10	10.4	
University of applied sciences	29	11.2	17	10.4	12	12.4	
University	63	24.4	33	20.4	30	31.3	
Not specified	17		11		6		
Age of first sexual intercourse (mean)	17.8 ± 3.2	[12–41]	18.2 ± 3.6	[13–41]	17.2 ± 2.2	[12–22]	**0.01**
Anal Intercourse							
Yes	97	35.9	51	30.2	46	45.5	**0.01**
No	173	64.1	118	69.8	55	54.5	
Not specified	5		4		1		
Previous anorectal disease							
Yes	84	31.5	53	31.5	31	31.3	0.97
No	183	68.5	115	68.5	68	68.7	
Not specified	8		5		3		
HPV vaccinated							
Yes	25	9.6	12	7.3	13	13.5	0.10
No	236	90.4	153	92.7	83	86.5	
Not specified	14		8		6		
Having ever heard of anal cancer							
Yes	118	43.7	78	46.2	40	39.6	0.29
No	152	56.3	91	53.8	61	60.4	
Not specified	5		4		1		
Condom used							
Yes	49	18.4	33	19.9	16	16.0	0.43
No	217	81.6	133	80.1	84	84.0	
Not specified	9		7		2		
Number of sexual partners							
0	3	1.1	2	1.2	1	1.0	**<0.0001**
1	54	20.4	45	27.1	9	9.1	
2–4	81	30.6	59	35.5	22	22.2	
5–10	68	25.7	34	20.5	34	34.3	
>10	59	22.3	26	15.7	33	33.3	
Not specified	10		7		3		
Smoking							
Yes	62	23.6	25	15.2	37	37.4	**<0.0001**
No	201	76.4	139	84.8	62	62.6	
Not specified	12		9		3		

HR-HPV: high-risk human papillomavirus (genotypes 16-18-31-33-45-52-58). Significant *p*-value < 0.05 are in bold characters.

**Table 2 cancers-14-05096-t002:** Univariate analysis of cytopathologic factors associated with cervical HR-HPV: PCR genotyping, cytology, pathology, and blood sample.

Predictive Variable	Total Population N = 275		Cervical HR-HPV-Negative N = 173		Cervical HR-HPV-Positive N = 102		*p*
	N	[%]	n	[%]	n	[%]	
HIV status							
Yes	96	35.8	64	37.9	32	32.3	0.36
No	172	64.2	105	62.1	67	67.7	
Not specified	7		4		3		
Anal HPV PCR							
Anal LR HPV							
Yes	76	31.9	38	25.9	38	41.8	**0.01**
No	162	68.1	109	74.1	53	58.2	
Not specified	37		26		11		
Anal HR HPV							
Yes	91	38.2	37	25.2	54	59.3	**<0.0001**
No	147	61.8	110	74.8	37	40.7	
Not specified	37		26		11		
Anal pathology							
Anal cytology							
Normal	218	87.9	146	93.0	72	79.1	**0.01**
LSIL	12	4.8	5	3.2	7	7.7	
HSIL	6	2.4	3	1.9	3	3.3	
ASCUS	10	4.0	3	1.9	7	7.7	
ASC-H	2	0.8	0	0.0	2	2.2	
Not specified	27		16		11		
Anal biopsy performed							
Yes	18	6.5	12	6.9	6	5.9	0.73
No	257	93.5	161	93.1	96	94.1	
Anal biopsy							
Normal	8/18	44.4	5/12	41.7	3/6	50.0	0.93
LSIL	7/18	38.9	5/12	41.7	2/6	33.3	
HSIL	3/18	16.7	2/12	16.7	1/6	16.7	
Cervical HPV PCR							
Cervical LR HPV							
Yes	56	20.4	17	9.8	39	38.2	**<0.0001**
No	219	79.6	156	90.2	63	61.8	
Cervical pathology							
Cervical biopsy							
Yes	81	29.5	35	20.2	46	45.1	**<0.0001**
No	194	70.5	138	79.8	56	54.9	
Cervical histology							
Normal	24/81	29.6	18/35	51.4	6/46	13.0	**<0.0001**
LSIL	14/81	17.3	8/35	22.9	6/46	13.0	
HSIL	43/81	53.1	9/35	25.7	34/46	73.9	
Cervical cytology							
Normal	181	72.1	127	80.9	54	57.4	**<0.0001**
LSIL	27	10.8	8	5.1	19	20.2	
HSIL	10	4.0	3	1.9	7	7.4	
ASCUS	24	9.6	16	10.2	8	8.5	
ASC-H	9	3.6	3	1.9	6	6.4	
Not specified	24		16		8		
Abnormal	10	3.6	4	2.3	6	5.9	

HR-HPV: high-risk human papillomavirus (genotypes 16-18-31-33-45-52-58). LR-HPV: low-risk human papillomavirus. PCR: polymerase chain reaction. LSIL: low-grade squamous intraepithelial lesion. HSIL: high-grade squamous intraepithelial lesion. ASCUS: atypical squamous cells of undetermined significance. ASC-H: atypical squamous cells cannot exclude high-grade squamous intraepithelial lesion. Significant *p*-values < 0.05 are in bold characters.

**Table 3 cancers-14-05096-t003:** Correlation between cervical and anal HPV status.

	Cervical HPV Status							
		Total N = 275		Negative N = 156		HR (±LR) N = 102		*p*
		n	[%]	n	[%]	n	[%]	
Anal HPV status	Negative	118	49.4	88	66.2	25	27.5	**<0.0001**
	HR (±LR)	91	38.1	31	23.3	54	59.3	
	Not specified	36		23		11		

HPV: human papillomavirus. HR-HPV: high-risk human papillomavirus (genotypes 16-18-31-33-45-52-58). LR-HPV: low-risk human papillomavirus. Significant *p*-values < 0.05 are in bold characters.

## Data Availability

The data presented in this study are available on request from the corresponding author. The data are not publicly available due to ethical comity restrictions.

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
