# Peer review of "Women with Cervical High-Risk Human Papillomavirus: Be Aware of Your Anus! The ANGY Cross-Sectional Clinical Study"

_cancers, 2022, doi:10.3390/cancers14205096_

Round 1

Reviewer 1 Report

The article titled “Women with cervical high-risk human papillomavirus be two aware of your anus! The ANGY cross-sectional clinical study” aimed to determine if patients with cervical HR-HPV infection had a concomitant infection of the anus with HR-HPV as well. 

Although the data presented in. this study are interesting, some obstacles prevent the manuscript publication in its current form.

These include:

In the Materials and method section, notification describing the Ethical committee approval(with its number in parenthesis)  and patients informed consent should be included in the text.

Although the starting authors intended to include 360 patients, the manuscript should be oriented on the number of participants included in the study. Thus, the power of the study should be described regarding the 275 participants whose data were analyzed in the study.

The univariate and multivariate analysis performed in the study should be identified/named in the materials and method section. 

Also, the covariates used for the multivariate analysis should be precisely described based on the type of analysis (dependent variable) for which they are used, i.e., escaping the term “such as.” Furthermore, the p-value designated as statistically significant should also be stated in the materials and method section (statistical analysis).

There is no need for an extensive description of numerical data (in the manuscript text) already presented in the tabular format.

The data presented in table 2 should be optimized- e.g., no range data or mean ± SD are shown; thus, no need for their mention in the uppermost row of the table.

Although common knowledge, the full names for the  HR (± LR) should be presented in the table legend. Also, symbols such as OR and CI should be described by their full name when shown for the first time in the manuscript text.

Some statements in the discussion section were not accompanied by corresponding reference(s).

A major revision of the manuscript is recommended.

Author Response

Reply to Reviewers 1
The article titled “Women with cervical high-risk human papillomavirus be two aware of your anus! The ANGY cross-sectional clinical study” aimed to determine if patients with cervical HRHPV infection had a concomitant infection of the anus with HR-HPV as well. Although the data presented in. this study are interesting, some obstacles prevent the manuscript publication in its current form. These include:
1. In the Materials and method section, notification describing the Ethical committee approval (with its number in parenthesis) and patients informed consent should be included in the text.
Reply from author: Dear reviewer, thank you for your suggestion. We modified accordingly in line 110.
2. Although the starting authors intended to include 360 patients, the manuscript should be oriented on the number of participants included in the study. Thus, the power of the study should be described regarding the 275 participants whose data were analyzed in the study.Reply from author:
Dear reviewer, thank you for pointing this out.
Indeed, the statistics have been performed on 275 participants instead of 360. The power of the study itself was determined « a priori » at the stage of design of the study, based on the hypothesis that histological abnormalities would be rare events, especially anal HPV-HR incidence would be low.
However, the prevalence of cervical HR-HPV was surprisingly higher than expected in our population study. In addition, the prevalence of anal HR-HPV was significantly different between cervical HR-HPV negative women (25.2%) and cervical HR-HPV positive women (59.3%), p<0.0001.
To highlight a such difference, with an alpha-risk of 5%, a power of 90% and a similar population ratio (173/102), the arcsin test reveals that 58 and 33 patients would be required, respectively in the cervical HR-HPV negative women group and the cervical HR-HPV positive women group.
The number of patients included in both groups was higher in our study, allowing sufficient statistical power to highlight such difference.
3. The univariate and multivariate analysis performed in the study should be identified/named in the materials and method section. Also, the covariates used for the multivariate analysis should be precisely described based on the type of analysis (dependent variable) for which they are used, i.e., escaping the term “such as.” Furthermore, the p-value designated as statistically significant should also be stated in the materials and method section (statistical analysis).
Reply from author:
Thank you for this pertinent comment. The paragraph on statistical analysis (line 118 onwards) was modified as following:
Page -3
“Patients were divided into two groups according to their cervical HR-HPV: positive or negative.
The chi-squared test was applied to compare qualitative variables that were expressed as n (percent). The ANOVA test was used to compare quantitative variables that were reported as mean ± standard deviation (SD). P values lower than 0.05 were considered as significant.
Variables yielding p values lower than 0.05 by univariate analysis were entered into a multivariate logistic regression model to determine variables independently associated with the presence of cervical HR-HPV. All statistical analyses were carried out using XLstat Biomed software (AddInsoft V19.4, Paris, France).”
4. There is no need for an extensive description of numerical data (in the manuscript text) already presented in the tabular format.
Reply from author: Dear reviewer, thank you for your suggestion to remove from the text the data described in the respective tables. It will help to improve reading fluency. We therefore removed text results of table 1 and table 2 at line 130 onwards and 160 onwards.
5. The data presented in table 2 should be optimized- e.g., no range data or mean ± SD are shown; thus, no need for their mention in the uppermost row of the table.
Reply from author: Dear reviewer, thank you for your comment. We deleted the “range” and “mean+/- sd” in table 2.
6. Although common knowledge, the full names for the HR (± LR) should be presented in the
table legend. Also, symbols such as OR and CI should be described by their full name when shown for the first time in the manuscript text.
Reply from author: Dear reviewer, thank you for your suggestion. We added the numbers of the HR HPV screened (16-18-31-33-45-52-58) in the table legends. (Line 158 and 229). A short explanation was also included in the Method section line 92. We proceeded the same for OR and CI in the manuscript text.
7. Some statements in the discussion section were not accompanied by corresponding reference(s).
Reply from author: Dear Reviewer, thank you for your comment. We went through the discussion, and added the references when appropriate, such as at line 232, 252
8. A major revision of the manuscript is recommended.
Reply from author: Dear reviewer, we proceeded to a major revision the manuscript, especially regarding the statistics and the English phrasing. We hope the present manuscript will match your comments and review, and we remain at your entire disposal for any queries or questions.

Reviewer 2 Report

In this article, Jacot-Guillarmod and colleagues evaluated if there is a correlation between cervical and anal HPV infections in a cohort of 275 women recruited in a university hospital in Switzerland.

Moreover, authors emphasize that the results of this study underline the importance to screen women for anal high-risk (HR)-HPV when diagnosing cervical HR-HPV.

It is important to well define if there could be an association between HR-HPV cervical and anal infection to improve prevention programs.

The aim of the study is clear, and results are promising but methods should be better described. The use of the English language in adequate.

This article is interesting; however, major revisions should be made to improve the manuscript.

Introduction: The introduction should include a section describing data already reported in literature about concomitant cervical and anal HPV infections.

Material and Methods: How was samples collection (both for cervical and anal samples) performed?

Authors should better describe which HPV test was used. Is it a molecular test? Which method for nucleic acids extraction was used?

In the results are reported high-risk and low-risk (LR) HPV positivity but how many HR and LR genotypes were considered to determine the HPV positivity?

It is important to describe methods used for HPV detection because different protocols could give different HPV positivity rates.

 Results: Has authors considered the possibility that women could have multiple HPV genotypes infection? A section regarding multiple HPV infections should be added in the Results.

Author Response

Reply to Reviewers 2

In this article, Jacot-Guillarmod and colleagues evaluated if there is a correlation between cervical and anal HPV infections in a cohort of 275 women recruited in a university hospital in Switzerland. Moreover, authors emphasize that the results of this study underline the importance to screen women for anal high-risk (HR)-HPV when diagnosing cervical HR-HPV. It is important to well define if there could be an association between HR-HPV cervical and anal infection to improve prevention programs. The aim of the study is clear, and results are promising but methods should be better described. The use of the English language in adequate. This article is interesting; however, major revisions should be made to improve the manuscript.

  1. Introduction: The introduction should include a section describing data already reported in literature about concomitant cervical and anal HPV infections.

Reply from author: Dear reviewer, thank you for your suggestion. We included a subsection briefly describing the concomitant cervical and anal HPV infection from line 59 onwards. “The same HR-HPV (such as HPV16 or HPV18) that play a major role in cervical cancer are also associated with severe diseases of the anal sphere. The concomitant HPV infection of the cervix and the anogenital region has been partially studied in some populations such as Hawaiian women, who had concurrent anal and cervical HPV infections in 13% of cases. (3, 4)”

  1. Material and Methods: How was samples collection (both for cervical and anal samples) performed?

Reply from author: Dear reviewer, thank you for your question. We added a description of the sampling in the Method section, Line 89 onward.

“The operation was completed during a single appointment. A cervical cytology smear (for PAP-test and HPV test), with biopsies if needed (according to the international standards of care in colposcopy), was conducted after vaginal examination and colposcopy on a gynecological chair. The proctology team conducted a proctology examination with high resolution anoscopy (HRA), along with cytological smears of the anus and the anal canal (with biopsies if lesions were visualized). Using a brush, ectocervical, endocervical, and anal samples were performed.”

  1. Authors should better describe which HPV test was used. Is it a molecular test? Which method for nucleic acids extraction was used?

Reply from author: Dear reviewer, thank you for your suggestion. We added a description of the HPV testing modalities in the Method section, Line 96 onward.

“Analysis of HPV test were performed with the MagNaPure 96 Total Nucleic Acids kit (Roche, Basel, Switzerland), with genotyping of 28 HPV genotypes (6, 11, 16, 18, 26, 31, 33, 35, 39, 40, 42–45, 51–54, 56, 58, 59, 61, 66, 68–70, 73, and 82) by the Anyplex™ II HPV28 kit (Seegene, Seoul, South Korea).”

  1. In the results are reported high-risk and low-risk (LR) HPV positivity but how many HR and LR genotypes were considered to determine the HPV positivity?

Reply from author: Dear Reviewer, regarding your question about number of HPV genotypes considered, we considered positivity to HPV when at least one HPV were found, either high risk or low risk among the 28 HPV genotypes associated with the Anyplex test. We added a statement at line 120.

  1. It is important to describe methods used for HPV detection because different protocols could give different HPV positivity rates.

Reply from author: Dear reviewer, according to your comment, and in line with question n°3, we added in the Method section a description of the HPV detection methodology. We confirm that screening methods were not changed during the study period.

  1. Results: Has authors considered the possibility that women could have multiple HPV genotypes infection? A section regarding multiple HPV infections should be added in the Results.

Reply from author: Dear reviewer, Thank you for your comment. Indeed, we could observe that women had multiple combinations of HPV, either HR or LR, from both sites (cervix or anus).

Among the 102 patients with cervical HR-HPV, 48 different combinations of HR-HPV were identified. The most frequent case was the exclusive presence of HPV 16.

Among the 56 patients with cervical LR-HPV, 21 different combinations of LR-HPV were identified. The most frequent case was the exclusive presence of HPV 42.

It seemed to us inappropriate to specifically analyze each combination. We tried to depict in the best possible way the correlation between anal and cervical HPV in Table 3.

Dear reviewer, we proceeded to a major revision the manuscript, with emphasis on the statistics and the English phrasing. We hope the present manuscript will match your comments and review, and we remain at your entire disposal for any queries or questions.

Round 2

Reviewer 1 Report

The authors have successfully revised the manuscript.

I have no further questions or remarks.